# Community Response to Burn Injuries: Examples from Dhading District of Nepal

**Bimal Singh Bist** [1,*]**, Bhagabati Sedain** [2] **, Maheshwar Shrestha** [3] **and Prativa Tripathi** [4]

1 Department of Health, Graduate School of Education, Tribhuvan University, Kathmandu 44613, Nepal
2 Department of Population Studies, Padmakanya Multiple Campus, Tribhuvan University, Kathmandu 44600, Nepal; bssedhai@gmail.com
3 Provincial Health Training Center, Ministry of Social Development, Butwol 32907, Nepal; maheshwarshrestha22@gmail.com
4 Nepal Medics, Kathmandu 44614, Nepal; prativa@nepalmedics.org
* Correspondence: bimalbist@gmail.com

**Abstract:** Burns are one of the most serious global public health challenges, and Nepal is no exception. This study aims to present national and local-level data regarding burn injuries within Nepal. Similarly, this study shows how the trained rural first responders respond to burn injuries at the community level, with an example from the Dhading district of Nepal. Police and Emergency Medical Services (EMS) records were used to describe the national and community-level burn injury patterns. The most common cause of burns was found to be household fire, mainly from cooking. The burn cases are distributed across all ages; however, young age group comprises a notable proportion. Victims who were injured but were still able to move primarily accessed emergency health services by walking to the closest facility. Mainly, burn victims received a dressing and cold sponging service at the primary health center. This study described the Emergency Medical Services (EMS) in detail and identified that appropriate training to the community people to respond the burns injuries minimizes the severity of the cases. Lessons learned from this project can be utilized to implement emergency burn injury management for the public and local responders in other rural areas at minimum costs. We recommend establishing burn care instruction in all rural/remote villages and health care centers in Nepal.

**Keywords:** burn; injuries; community responders; first responders; Emergency Medical Services

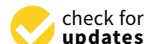



## 1. Background

Burns are a major public health problem in low-income countries and the leading cause of death and disability [1,2]. Vidal-Trecan et al. revealed that burns were more frequent in rural areas than the urban [3]. World Health Organization's (WHO) 'Burns Facts' show that low-and middle-income populations are at the greater risk of burns [4]. In Nepal, the largest proportion of the population (82.9%) resides in rural communities [4,5] with poor economy [6]. Burns are the second most common injury in rural Nepal, leading to 5% of disability [7]. People are uninformed about the treatment of burns; most of the injured are often brought to the hospital only after wounds have started to smell [6,8]. A systematic review of the epidemiology of burn injuries in Nepal shows that due to the long-elapsed time between burn injury and hospital treatment, larger burn-related morbidity and mortality resulted [9].

Most burn injuries occur within or around the home environment [7,8]; however, the national-level information on the magnitude of burn injuries is not available [9]. A study based on the Global Burden of Disease study identified that fire/burns represent 0.2% (0.14% to 0.29%) of the total deaths in Nepal [5,10]. Similarly, the World Health Organization estimates 184 fatal burns (with 58% females) for the year 2019 in Nepal [11].

Traditionally, people use home treatments for the burn injuries in rural Nepal. Use of domestic herbs, honey, raw eggs, wet ash, potato, tomato, and toothpaste on burn wounds as first aid is common [8]. Visiting a health facility for the burn treatment is often a secondary choice for rural people due to absenteeism of health workers at the primary health centers, lack of resources, and a shortage of required medicines might be a discouraging factor for rural people to go to health post [12]. Lack of adequate epidemiological data at the local health posts challenged the researcher to identify the actual injury causes and burdens from burns in Nepal [9,10]. However, some moderate to severe burn injuries are referred from the community-based health centers to feeders to tertiary hospitals and some studies are published based on those hospital-based burn information [12–14].

Communities in rural areas are scattered, and health facilities are accessible for 1–2 hours' walking distance. Although a study shows that 61.8% of Nepali households have access to health facilities within 30 min, there is a significant urban (85.9%) and rural (59%) discrepancy [15]. Yet, in rural Dhading, health facilities may be even further, up to 2–5 h walking distance. But we have experienced an incident that the injured have waited for a couple of days to access referral facilities. With limited road access to many rural villages, bamboo baskets, stretchers, and being carried on someone's back are commonly the means of transportation for burn victim from rural communities to health facilities.

This study aims to present the national-level and local-level information about burn injuries and to describe how the trained rural first responders respond to burn injuries at the community level, with an example from the Dhading district in Nepal.

## 2. Materials and Methods

After the devastating earthquakes in 2015, national and international experts recommended establishing Emergency Medical Services (EMS) [16]. Based on the recommendations, the Ministry of Health and Population implemented a project in one of the most earthquake-affected areas, Dhading district. The project aimed to prepare a rural and remote village responders for immediate response to the acutely ill and injured. EMS is an ongoing project in Dhading district, in which a rural-based first responding model, namely "Rural First Responder (RFR)," was implemented. The Rural First Responder refers to a non-medical professional, community member who participated in a three-day first aid training course [14]. This project was successful in training about 900 RFR during 2016–2018. An Emergency Communication Center (ECC) collected data on types and treatment of injuries for thirty months. This study used only the burn injury information and extracted possible variables for descriptive analysis.

Regarding the national-level data on burn deaths and injury, this study utilized Nepal Police's burn incidents records analysis for the year 2019. The analysis was published in the research report conducted under Small RDI Grant (for faculty members) supported by University Grant Commission (UGC), Nepal [17]. This study also presents how the locally trained RFR responds to burn injury victims after receiving their training. Similarly, we have made several recommendations for decision-makers to develop a burn-related first response system in rural areas.

## 3. Results

### 3.1. Situation of Burn Injuries and Deaths in Nepal

According to Nepal Police Daily Incident Reporting System (DIRS) records, more than 2500 incidents of fire occurred in 2019, in which 88 people died and 179 were injured. This resulted in a huge loss of national economy from property damage and lost productivity. Among the different age categories, the 20–29 and 30–39 age group has a larger number of burn injuries (Figure 1). Further, Figure 1 reveals that a larger number of children below 10 years old suffered from fatal burns and injuries.

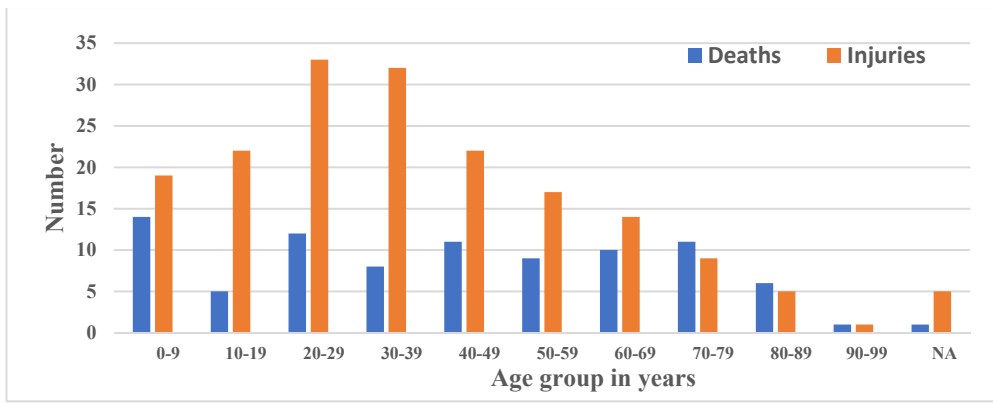

**Figure 1.** Reported deaths and injuries from burns by age group, Nepal, 2019 [17].

For all injuries except for burns, the male population outnumbers females [2,10,11]. In the case of burns, 60% were females, among a total 88 fatal cases and 179 nonfatal burns. The police records analysis also showed that the number of burn deaths and injuries were higher in the Terai region of Nepal, followed by the Hilly and Mountain region [17].

*3.2. Emergency Medical Services (EMS)*

Under this EMS project, various activities have been conducted. The major activities were; selection of RFR, training for RFR, data capturing system development, service delivery, evaluation of burn injuries by RFR, development of Rural First Responders model. The authors, BSB, MS, and PT, were directly involved to develop the training package and implementing the project in Dhading district.

3.2.1. Training for RFR

The RFR Training curriculum was a 24-h course. The course covered the introduction of RFR, patient assessment, cardiopulmonary resuscitation information on different kinds of injuries and pre-and-post-test (Box 1). During the training, RFRs were also informed about whom to communicate to both during the incident (to assist with proper transportation of the injured) and after (Call-in follow up calls) for the purpose of data collection. The details are presented in Figure 2.

**Box 1.** RFR Training curriculum.

---

**Day 1, 8 h**
Lesson 1: Introduction, Objectives, Emergency Communication Center, Pre-test, First Aid, RFR, EMS, Patients' communication, and Roles and Responsibilities of RFR.
Lesson 2: Patient Assessment System, Hands-on Approach, and Practical Skills.
Lesson 3: Life Threats, triage, hemorrhage control, shock, Airway issues, Rescue Breathing, CPR, and day conclusion, Life Threats, Open Pneumothorax, Tension Pneumothorax, and Seizures.
**Day 2, 8 h**
Lesson 4: Musculoskeletal Injury, Head Injury, Spinal Injury, Muscle & Ligament Injury, Bone, Burns, Blisters, Crush Injury, Eye Injuries, and Dental Emergencies.
Lesson 5: Medical Illness, Cardiac Illness, Stroke, Respiratory Illness, Abdominal Problems, Constipation, Diarrhea, Nausea & Vomiting, Allergic, Reaction, Diabetes, Hypoglycemia & Hypoglycemia, Scenario, and Day's Conclusion.
**Day 3, 8 h**
Lesson 6: Environmental Topics, Heat Illness, Hypothermia, Frostbite, High Altitude Sickness, Poisoning, Bites (Snake, animal bites), Stings, Lightening, Drowning.
Lesson 7: Childbirth.
Lesson 8: Evacuation Techniques
Practical Skill, Post-test, Result, and Certification, Training Evaluation, and Graduation Ceremony
Local resources used: RFR were taught how to use local resources, such as clothes, clean water, plastic bags, sticks, and tarpaulin.

---

### 3.2.2. Selection Criteria of RFR

The District Health Office, Dhading set the criterion to select the trainees of the 24-h RFR course [18]. The criterion were: age between 15–55 years, local residents, active in the community, readily available, accepted by the community, serving heart "not to be served but to serve others," independent, willing to learn, participation of 50% male, 50% female, able to attend three days training, and literate.

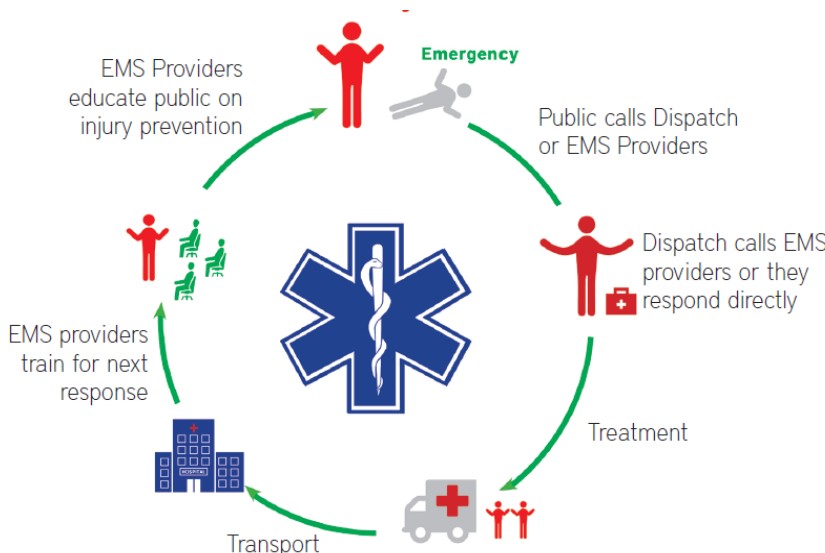

**Figure 2.** Emergency Medical Services Model of RFR. Source: Rural First Responder Handbook, 2016 [19].

### 3.2.3. Data Capturing System

The training organizer provided contact details, i.e., mobile number of Emergency Communication Center (ECC), to teach the trainees to practice medical communication process. The dispatchers stationed at Emergency Communication Center (ECC) gathered daily updates from the five trained responders through check-in calls. Similarly, the RFRs called the ECC to inform them about the incidents the RFR responded to in the community.

### 3.2.4. Service Delivery System

Service delivery includes community engagement with four components: embeddedness (traditions, ethics, norms and values); institutional process (execution, legislation, judiciary, and organizational rules and regulations); governing system (management structures); and resources management [19]. During the project period, the in-charge of the Primary Health Center of the respective community (Doctors and Paramedics) also participated in a ten day 'Rural Medical Responders' (RMR) training course to provide support to RFRs. RFR course graduates were provided EMS supplies, such as Cravats (triangular bandage), 4–6" Elastic wrap (Extensible Bandy), Adhesive Tape, Scissors, Forceps, 3 × 4 non-stick gauze Pads, 4 × 4 Gauze Pads, Gloves Pairs, Black Board marker, Alcohol swabs, 3" conforming roll gauze (cotton roll), 50 cc+ irrigation syringe, Plastic garbage bag, safety pins, and oral rehydration salts. RFRs were directed to refill their supplies at their local health respective health facilities.

### 3.2.5. Rural First Responder Evacuation of Burn Victim

A dispatcher located in the ECC at Dhading District Hospital provides, 24/7, dispatches responders to the scene for evacuating acutely ill and injured. The ECC Dispatch Center, Dispatcher, RFRs, Ambulance service providers, and health service providers work jointly at the local level to improve service delivery. For a mass casualty, a team of trained

RFR and RMRs located at neighboring villages are dispatched to triage and evacuate the injured.

### 3.3. Rural First Responders Model Response for Burn Injuries

Ministry of Health and Population implemented Emergency Medical Services (EMS) project to train community responders to stabilize and provide first aid services to acutely ill and injured during any disaster and post-earthquake of 2015. Ambulance Drivers, Business person, Farmers, Female Community Health Volunteers (FCHV), Health workers, Mother groups, Students, Teachers, Traffic police, and general public are intended to serve the community as Rural First Responders.

Daily allowances for travel, food, and meeting was provided to those who participate in training and development-related activities. But RFR is a voluntary work to help neighbors during health emergencies. Still, countable trained responders were less motivated due to no provision of allowances. This was the biggest challenge for the EMS project in Dhading. The primary tasks of the project were to develop a handbook for Rural First Responders, lesson plans for facilitators, preparing community-based rural first responders, and establishing an emergency communication center at the local level.

The following EMS model (Figure 2) describes how the Emergency Medical System works at the local level [20]. If an incident occurs, the villagers contact their local RFR and request an immediate response. The RFR responds and provides first aid. In the case where patient requires further referral, he/she is transported to the health care facilities. After referring the person, RFR evaluates the overall process for further improvement in the system.

### 3.4. Burn Case Analysis for Project District Dhading

During a 30-month period (February 2017 to July 2019), this project collected information on 404 burn cases. Among the cases, RFRs responded to 275 burn cases, and Female Community Health Volunteers (FCHVs) responded to 129 burn cases. In the course of the study period, about 15% were responded to by the health workers and the rest by the non-health workers. In addition, this project identified that an overwhelming majority (90%) of the burn victims accessed health facilities by walking. Further, it found that about 96% of burn injuries occurred at home.

Figure 3 shows the percentage distribution of means of transportation used to respond burn victims in Dhading. The figure reveals that the majority of injured people (90%) arrived at local health facility by walking, with the remaining 10% being transported by hand-carried stretcher, ambulance, local jeep, and local truck.

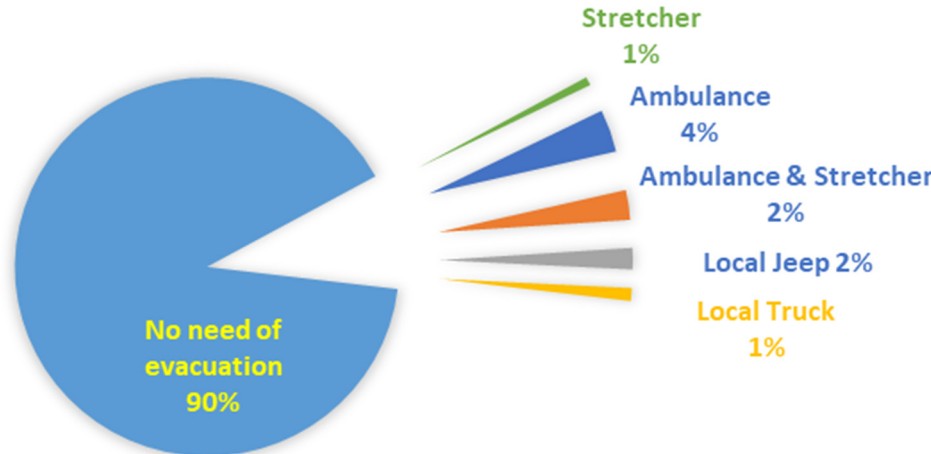

**Figure 3.** Distribution of means of transportation used to respond burn victims in Dhading.

Figure 4 shows the causes of burn injuries recorded in the emergency medical services system. Among the total cases, cooking caused 70% of the burn injuries, followed by heating during cold season. In a rural community, 15% were injured by firewood; people often use a kerosene lamp for light, reflecting data that 4% were injured from the flame burn. Similarly, 9% were associated with an electrical burn, and only 2% of burn injuries was caused by jungle fire.

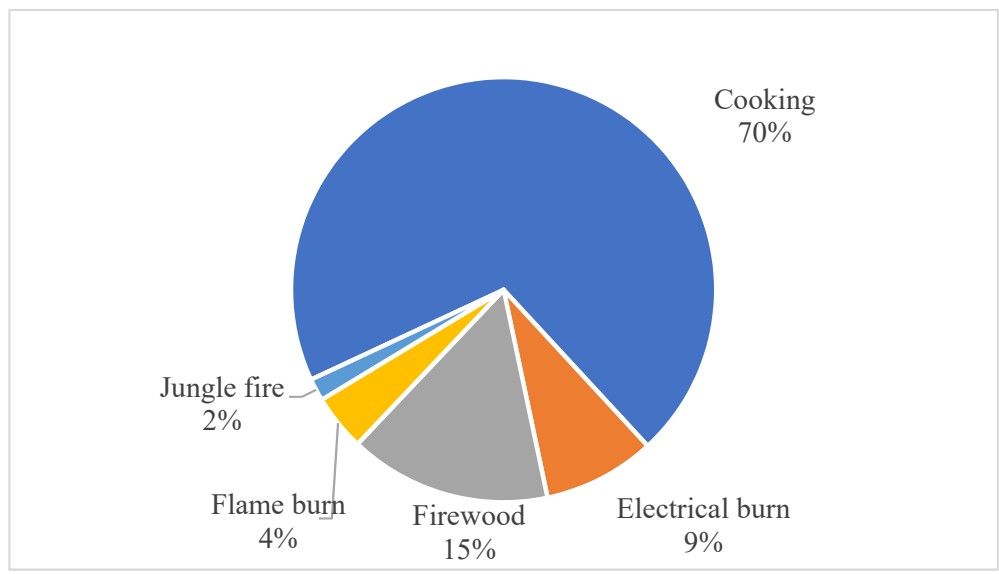

**Figure 4.** Distribution of causes of burn injuries in Dhading.

Figure 5 shows the type of treatments provided by RFR to the burn victims. About three-fourths (76%) of the victims received dressing services from the RFR, followed by cold sponging (20%).

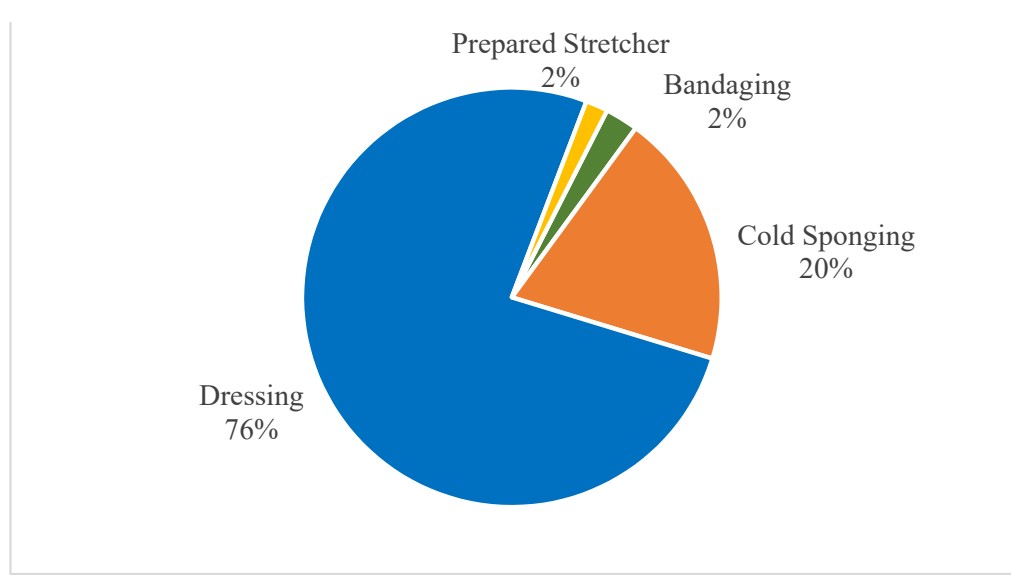

**Figure 5.** Distribution of type of treatments provided by RFRs in Dhading.

*3.5. Case Studies*

Box 2 represents how the trained Rural First Responder responded to the burn injuries.

**Box 2.** Case story-1.

> On 8 April 2017, at around 9:00 a.m., a 70-year-old woman was cooking food for lunch. Suddenly, the utensils with which she was cooking the vegetable curry became imbalanced and hot curry spilled over her body. The burn was severe, and her family members first poured drinking water and applied fresh Aloe Vera gel in the wound.
>
> While this incident occurred, a trained Rural First Responder, Maiya (name changed), was on her regular teaching job in a school. When she came to know about the incident from a villager, she immediately called to Emergency Communication Center in District Hospital, and RFR was suggested to evacuate her by ambulance; and, in two hours, she was transported to Teaching Hospital at Kathmandu (70 km far from the place incident) via ambulance, for which she was charged $60.
>
> After being admitted for three months and spending around $1000 (direct costs including medicines, as reported by the patient's family members), Malati was discharged from the hospital.

These examples revealed that a proper on-time response at community level could help to save lives from burn injuries.

## 4. Conclusions

Burns have caused notable deaths and injuries in Nepal. The majority of the burn injuries occurred inside the house, mainly while cooking. Most of the burn injuries occurred in the economically active age group (20–49 years). So, increased efforts in awareness and response would likely lead to a significant reduction of burn-related death and disability in the community. Proper and timely response for burns can reduce morbidity and mortality. This study identified that the appropriate training to the community people to respond to burn injuries minimizes the severity of the cases. We can implement a burn injury management system in rural areas with minimum costs, and this EMS Model can also be applied to other types of injuries. Further, this study shows how medical records can maintain from the local health institutions through EMS. Therefore, it is recommended to operate EMS model in all health care centers in Nepal for immediate response to the people in the community and to establish data recording system at the local level.

**Author Contributions:** Conceptualization: B.S.B. and B.S.; Methodology: B.S., M.S. and B.S.B.; Data management: M.S., P.T., B.S.B. and B.S.; Writing—original draft preparation: B.S.B., B.S.; Review and editing: B.S. All authors have read and agreed to the published version of the manuscript.

**Funding:** This research received no external funding.

**Institutional Review Board Statement:** Not applicable.

**Informed Consent Statement:** Not applicable.

**Data Availability Statement:** This study did not report any primary data.

**Conflicts of Interest:** The authors declare no conflict of interest.

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
