# Peer review of "Community Response to Burn Injuries: Examples from Dhading District of Nepal"

_2673-1991, doi:10.3390/ebj2030005_

Round 1
Reviewer 1 Report
Excellent paper informing that burns is a leading cause of death and disabilities in a low income countries. Burn is the second most common injury in rural Nepal, leading to 5% disability. Burns occurs in small single room houses and most of the patients are brought late and infected to the hospital. Primary health center lack of resources, medicine and absenteeism is a problem. In rural Dhading access to health take 2-5 hours walking.
This ongoing project will rural first responders may help to reduce burn morbility and mortality, and can be an example for other low income countries.
I will revise the Figure 1: Reported death and injuries..... (Bar 30-29 is wrong, it can be taken out).
Author Response
Response 2: Thank you so much for this suggestion. We have changed the figure.
Reviewer 2 Report
New source of data to describe the occurrence of injuries in rural area, such as police reports, could contribute to better understand the burden of burns in limited resource settings. Furthermore, the education of non-medical personnel in provision of first aid in case of burns could reduce the delay in reaching adequate care in rural zones.
However, this manuscript has several issues:
- The logical succession of the sentences in the manuscript is lacking, in particular in the introduction, which makes it difficult to follow the thread of the text.
- I´m wondering whether the following statements are evidence-based or based on anecdotes: “people use home treatment for the burn injuries in rural Nepal. People use items such as domestic herbs, honey, raw eggs, wet ashes, potato, tomato and tooth paste on burn wounds as first aid. Usually faith-driven family of the burn victims visit witch-doctor for healing. Visiting health facilities for the burn treatment is a secondary choice. Rural people wait for the possible remedies of burn injuries.” In the first case (evidence-based), I suggest adding the opportune references, in the second case (anecdotes), I think stories and cliché should lead to investigate if they are true and not be presented as facts.
- The aim as it is described is not clear to me. Was your plan to test the availability of new data sources, such as Police reports? Or to extrapolate from new data source epidemiological data of burns in the region?
You stated also that you planned to” describe community response to burn injuries with examples from Dhading district, Nepal”, but in results you just described how the “Rural First Responder” program was organized and reported just one example of community response, in one occasion.
- From the Methods as it is presented, I cannot understand the exact role of the authors in the “Rural First Responder” program. Were they organizer, did they contribute to develop this program? How was the program developed and by whom?
- Which was the study design? A retrospective data study based or an implementation study? Other?
- Which data did you analyze, and which was the exact data source for each data? Did you combine information extrapolated from emergency communication center, monthly reports, project evaluation report, police report, etc? How did you combine this data? Which statistical analysis did you use? Did you apply for ethical approval?
- Most of Results consisted in the description of how worked the “Rural First Responder” program, and again, I´m wondering how the authors contributed to this program. However, describing one case, as that in Box 2, is not enough to assess the effectiveness of this intervention and draw general conclusion.
Author Response
However, this manuscript has several issues:
1. The logical succession of the sentences in the manuscript is lacking, in particular in the introduction, which makes it difficult to follow the thread of the text.
Response 1: Thank you so much. We worked on the introduction section and revised it.
2. I´m wondering whether the following statements are evidence-based or based on anecdotes: “people use home treatment for the burn injuries in rural Nepal. People use items such as domestic herbs, honey, raw eggs, wet ashes, potato, tomato and tooth paste on burn wounds as first aid. Usually faith-driven family of the burn victims visit witch-doctor for healing. Visiting health facilities for the burn treatment is a secondary choice. Rural people wait for the possible remedies of burn injuries.” In the first case (evidence-based), I suggest adding the opportune references, in the second case (anecdotes), I think stories and cliché should lead to investigate if they are true and not be presented as facts.
Response: 2: Thank you so much, as per the suggestion, the reference has been reflected.
3. The aim as it is described is not clear to me. Was your plan to test the availability of new data sources, such as Police reports? Or to extrapolate from new data source epidemiological data of burns in the region? You stated also that you planned to” describe community response to burn injuries with examples from Dhading district, Nepal”, but in results you just described how the “Rural First Responder” program was organized and reported just one example of community response, in one occasion.
Response: 3: Now, the aims of the study clearly written as follow;
This study aims to present the national-level and local-level information about burn injuries and to describe how the trained rural first responders can respond to burning injuries at the community level, with an example from the Dhading district of Nepal.
4. From the Methods as it is presented, I cannot understand the exact role of the authors in the “Rural First Responder” program. Were they organizer, did they contribute to develop this program? How was the program developed and by whom?
Response: 4: Thank you for this suggestion. The authors, BSB, MS and PT were directly involved to develop the training package and implementing the project in Dhading district. We also contributed to explain how the “First Rural Responders” were trained and worked to respond to burn injury in the community. The program was developed by the Ministry of Health and Population. These responders were trained through the project named “Emergency Medical Services.”
5. Which was the study design? A retrospective data study based or an implementation study? Other?
Response: 5: Some of the evidence presented in the study is based on secondary information, and it is an implementation study.
6. Which data did you analyze, and which was the exact data source for each data? Did you combine information extrapolated from the emergency communication center, monthly reports, project evaluation report, police reports, etc? How did you combine this data? Which statistical analysis did you use? Did you apply for ethical approval?
Response: 6: This study used already published information for the national evidence on burn injury, and the study did not directly contact people related to the burn injuries: simple descriptive analysis and graphical representation used in the analysis.
7. Most of Results consisted in the description of how worked the “Rural First Responder” program, and again, I´m wondering how the authors contributed to this program. However, describing one case, like that in Box 2, is not enough to assess the effectiveness of this intervention and draw a general conclusion.
Response 7: Thank you. Now the article is revised and described “Rural First Responder” training and the modality of the “Emergency Medical Services” project. The authors (B.S.B., M.S., and P.T.) were directly involved in designing and implementing the program.

Reviewer 3 Report
I think the project and the associated manuscript is very important and good. The idea is great and should be followed in other low-income countries. There is no better help on the ground than to establish such a system.
I would still be interested to know what the challenges and difficulties were in training the RFR and if there was a drop out rate. Another question would be if it would not make sense to offer annual refreshers or continuing education.
Author Response
Point 1: I think the project and the associated manuscript is very important and good. The idea is great and should be followed in other low-income countries. There is no better help on the ground than to establish such a system. I would still be interested to know what the challenges and difficulties were in training the RFR and if there was a drop out rate. Another question would be if it would not make sense to offer annual refreshers or continuing education.
Response 1: Thank you so much for your kind words. Daily allowances for travel, food, and meeting are provided to those who participate in training and development-related activities. But RFR is voluntary work to help neighbors during health emergencies. Still, then, countable trained responders were less motivated due to no provision of allowances. This was the biggest challenge for the EMS project in Dhading.

Round 2
Reviewer 2 Report
Thank you for considering my suggestions and including them in your manuscript. I´m happy with the revision you made. The narrative flows well and, in the current draft, the manuscript is a useful guide for other settings that want adopting your implementation strategy and improve the first management of burns. I just suggest to add in Method, that it is a implementation study.
Author Response
Response to Reviewer 2 Comments
Point 1: Please revise your manuscript according to Reviewer#2's comments and
Upload the revised file within 5 days.
Response 1: Our manuscript has been edited by a native English speaker and resubmitted on 21 May 2021.
Point 2: Any revisions made to the manuscript should be marked up using the
“Track Changes” function if you are using MS Word/LaTeX, such that
Changes can be easily viewed by the editors and reviewers.
Response 2: We have used the track change function; the native English speaker had done editing work on a separate document, but I retyped all edits into the original manuscript. For your kind reference, I am happy to upload the document provided by a native English speaker.
Point 3: Please provide a short cover letter detailing your changes for the
Editors’ and referee’s approval.
Response 3: Cover letter submitted, thank you.
Point 4: If you are not a native English speaker, we recommend that you have
your manuscript professionally edited before submission or read by a native
English-speaking colleague. This can be carried out by MDPI's English editing
Service. More information can be found at this link:
https://www.mdpi.com/authors/english
Response 4: Our manuscript has been edited by a native English speaker, thank you.
